# Rapid Detection of *Salmonella* Typhimurium During Cell Attachment on Three Food-Contact Surfaces Using Long-Read Sequencing

**DOI:** 10.3390/microorganisms13030548

**Published:** 2025-02-28

**Authors:** Daniela Bermudez-Aguirre, Shannon Tilman, Joseph Uknalis, Brendan A. Niemira, Katrina L. Counihan

**Affiliations:** 1Characterization and Interventions for Foodborne Pathogens Research Unit, Eastern Regional Research Center, United States Department of Agriculture, Agricultural Research Service, 600 East Mermaid Lane, Wyndmoor, PA 19038, USA; daniela.bermudez@usda.gov (D.B.-A.); shannon.tilman@usda.gov (S.T.); brendan.niemira@usda.gov (B.A.N.); 2Microbial and Chemical Food Safety Research Unit, Eastern Regional Research Center, United States Department of Agriculture, Agricultural Research Service, 600 East Mermaid Lane, Wyndmoor, PA 19038, USA; joseph.uknalis@usda.gov

**Keywords:** *Salmonella*, foodborne pathogens, cell attachment, egg, biofilms, long-read whole genome sequencing, bioinformatics

## Abstract

*Salmonella* spp. are pathogenic microorganisms linked to foodborne outbreaks associated with eggs and egg products. *Salmonella* can resist sanitation of egg processing equipment and form biofilms on food-contact surfaces. A major challenge for controlling *Salmonella* is the ability to detect the cells during the early stages of attachment to indicate that interventions are needed to sanitize the surface. This research investigated the use of long-read sequencing to identify *Salmonella* during the early stages (0–5 h) of cell attachment to three common food-contact surfaces—stainless steel, silicone, and nylon—and compared it with traditional microbiological methods. Results of the conventional plate counts showed that the detection of *Salmonella* began after three hours of incubation, with less than 1 log CFU/cm^2^ of growth. Silicone had the highest number of *Salmonella* attached (0.87 log CFU/cm^2^), followed by stainless steel (0.70 log CFU/cm^2^). Long-read whole genome sequencing identified attached *Salmonella* on stainless steel, silicone, and nylon after only one hour of incubation. The results of this study suggest that long-read sequencing could be a very useful method for detecting *Salmonella* at low concentrations in the processing environment during the early stages of cell attachment to food-contact surfaces, allowing for correct sanitation intervention.

## 1. Introduction

Every year, approximately 153 million cases of gastroenteritis and 57,000 deaths are reported worldwide due to infections caused by *Salmonella* acquired from food [1,2]. Large economic losses also result from foodborne disease outbreaks, with an annual worldwide cost of 110 billion U.S. dollars, 17 billion of which correspond to the U.S. [3]. *Salmonella* is the primary foodborne pathogen of concern in the poultry and egg industries, and it is the second most common pathogen linked to foodborne outbreaks overall [2,4,5,6,7].

*Salmonella* has been widely reported on the surface of egg processing equipment, such as conveyor belts, as well as in floor drains and wash tanks [8]. Some microorganisms, including *Salmonella*, can adhere to food processing equipment surfaces and remain viable after the cleaning and sanitation procedures, leading to food cross-contamination [5,6]. Irreversible *Salmonella* attachment to surfaces can take from 20 min to 4 h, after which biofilm formation may occur [9]. Biofilms are bacterial aggregates that often contain more than a single microorganism. Bacteria are usually embedded in complex structures of self-produced extra polymeric substances (EPS), such as extracellular DNA, lipids, proteins, and polysaccharides [10,11]. *Salmonella* forms biofilms as a survival mechanism when exposed to severe physical and chemical conditions. One study collected samples from 15 locations in six commercial poultry processing plants at three time points and detected *Salmonella* on equipment after poultry processing (36%), after cleaning (12%), and after sanitation (9%) [6]. Cross-contamination of eggs or poultry with *Salmonella* due to contact with contaminated processing surfaces represents a food safety risk.

Bacterial biofilms are influenced by temperature, pH, nutrients, and serovar [11,12]. The molecular characterization of 25 *Salmonella* serovars from the poultry industry showed the presence of specific genes that regulate biofilm formation, sanitizer tolerance, and antibiotic resistance [5]. The genes involved in the production of extracellular components such as curli, fimbriae, and cellulose allow surface adhesion and biofilm production [12]. Temperature influences the production of these extracellular components. For example, at 22 °C, *Salmonella* produced 91% of these components, while at 37 °C, none were produced [12]. Additionally, neutral pH favors biofilm formation in most microorganisms. *Salmonella* formed stronger biofilms in pH 7 medium compared to pH 4.5 [13]. *Salmonella* has also been shown to respond to nutrient deprivation by forming biofilms [13].

The food-contact surface material—including stainless steel, glass, silicone, polyurethane, aluminum, Teflon, and wood [11,13]—influences the adhesion of microorganisms and biofilm formation [11]. Some properties related to the ability of microorganisms to adhere to materials include roughness, porosity, and contact angle [13,14]. For example, polyethylene terephthalate (PET), glass, and silicone have higher roughness than aluminum and Teflon. Meanwhile, the contact angle determines the wetting properties of the material. Hydrophobic materials, such as Teflon, have higher contact angle values, while hydrophilic materials, such as glass, have lower contact angles [13]. Each microorganism has unique adhesion properties that allow it to attach or detach in a different way to the food contact-surfaces.

Different interventions have been reported to remove biofilms from food-contact areas, such as sanitizers [15], cold plasma [16,17,18,19], and ultrasound [20,21]. However, one of the main challenges is the early and accurate detection of *Salmonella* on surfaces before biofilms are formed, so an effective intervention technology can be applied before bacterial contamination spreads to other areas or food products. Conventional microbiological methods, such as plating, are time-consuming, where results take at least 24–48 h, and the sensitivity to detect the presence of pathogens is limited. Faster detection methods for *Salmonella* include immunological detection techniques, nucleic acid detection, Raman spectroscopy, and biosensors [22]. These methods typically provide results in minutes to hours, although some immunological assays can take up to three days [23,24], and they have all shown the ability to detect *Salmonella* in various foods, including meat, produce, milk, and juice [22]. Biosensor and Raman spectroscopy have demonstrated high specificity through the use of antibodies and aptamers, but sensitivity can be very poor, with detection limits up to 10^8^ CFU mL^−1^ [24,25]. Additionally, samples for biosensor and Raman spectroscopy need to be carefully pretreated before testing, and these methods are not easily transferrable outside laboratory settings [25]. The specificity of immunological methods vary based on the antibody being used, and sensitivity ranges from 1–10^4^ CFU mL^−1^ [22,24]. Nucleic acid-based assays tend to have the highest specificity (93–100%) and the highest sensitivity (5–500 CFU mL^−1)^ [22,24]. However, a disadvantage of immunological and nucleic acid-based methods is that they are pathogen-specific, which may require multiple tests per sample depending on the bacteria being targeted [24].

Long-read whole genome sequencing technology has advanced significantly in the past decade and has the potential to reduce the time needed to identify foodborne pathogens compared to culture-based methods [3]. Nanopore-based sequencers are small and portable, which allows for testing outside of traditional laboratory settings. The long reads also facilitate genome assembly, and real-time analysis can generate results in hours instead of days [26,27]. Sequencing is also sensitive, and virulence genes from pathogenic bacteria have been detected in very low concentrations of DNA [3]. Long-read sequencing has not been used for the direct detection of foodborne pathogens on food-contact surfaces. Therefore, this research aimed to compare traditional culture-based methods and novel sequencing methods for identifying *Salmonella* during the early stages of cell attachment to three common materials found in the food industry: stainless steel, silicone, and nylon.

## 2. Materials and Methods

### 2.1. Microbial Strain

*Salmonella enterica* serotype Typhimurium 53,647 (American Type Culture Collection, Manassas, VA, USA) from the Eastern Regional Research Center Culture Collection was used to evaluate its attachment to different food-contact materials and to compare conventional and novel microbial detection methods. This strain was selected because of previous research conducted with eggs in a BSL-1 facility. The stock culture and preparation of the inoculum have been described in detail by Bermudez-Aguirre et al. (2025) [28]. Briefly, a thawed loopful from the frozen stock culture was grown overnight in Tryptic Soy Broth (TSB: MP Biomedicals, LLC., Solon, OH, USA) with 0.6% Yeast Extract (YE: Fisher Bioreagents, Fair Lane, NJ, USA) (TSBYE) and then transferred to fresh TSBYE for an 18-h enrichment. Centrifuged and washed cells were used to prepare fresh inoculum for the experiments. The concentration of *Salmonella* in the inoculum was confirmed for each experiment by plating dilutions on Xylose Lysine Deoxycholate (XLD; DifcoTM, Sparks, MD, USA) agar plates and incubating overnight. Black colonies confirmed the presence of *Salmonella* cells.

### 2.2. Food-Contact Surfaces

This study tested the early attachment of *S. typhimurium* to three common materials found in processing lines of the egg industry: stainless steel (304 L), silicone, and nylon (all food-grade). Coupons made from these materials were purchased from Biosurface Technologies Corp. (Bozeman, MT, USA). Each coupon measured 12.7 mm in diameter and 3.8 mm in thickness. Before each experiment, coupons were prepared using the following procedure: each coupon was washed with tap water and alkaline liquid detergent (Contrex, Decon Labs, Inc., King of Prussia, PA, USA) for a few minutes. Afterward, coupons were immersed in a 70% ethanol solution (*v*/*v*) at room temperature for one hour. The coupons were then rinsed with sterile deionized water and air-dried inside a biosafety cabinet for 15 min on each side. Coupons were wrapped in aluminum foil and sterilized in an autoclave at 121 °C for 30 min.

### 2.3. Cell Attachment and Quantification

Liquid whole egg (LWE) was used as the medium into which *Salmonella* was inoculated and incubated with the coupons. Commercial LWE was purchased from a local supermarket and kept under refrigerated conditions (4 °C) until use. The expiration date of each bottle of LWE was checked before purchasing, and the initial microbial quality of LWE was tested before each experiment using Aerobic Plate Count PetrifilmTM (3M Microbiology Products, St. Paul, MN, USA), incubated at 37 °C for 24–48 h. No viable cells were detected in any of the samples using this methodology. Samples of LWE were inoculated with 10^3^ CFU mL^−1^ *Salmonella* (chosen based on previous experiments) and stomached (400 Circulator Seward, West Sussex, UK) at 230 rpm for 3 min to homogenize the *Salmonella* and LWE.

Polystyrene culture plates (12-well, Corning, NY, USA) were prepared with 2 mL of inoculated LWE pipetted into each well, and one coupon per well was added using sterile tweezers. Plates were covered with lids and manually shaken to allow contact between the LWE and the full surface of the coupon. During each experiment, the three tested materials (stainless steel, silicone, and nylon) were exposed to the same inoculum and processing conditions (i.e., same incubator, same temperature). Culture plates were incubated inside a storage chamber (MIR-154-PA, Panasonic Healthcare Co., Ltd., Tokyo, Japan) at 37 °C under static conditions. Samples were removed at 1 h for sequencing and at 0, 1, 2, 3, 4, and 5 h for cell quantification.

### 2.4. Cell Attachment and Quantification Using Conventional Microbiological Methods

Each coupon was removed from the culture plate and transferred with sterile tweezers into 10 mL of sterile saline solution (0.85% *w*/*v*) for 1 min to remove planktonic cells. Each coupon was then transferred to 5 mL of fresh saline solution containing glass beads and vortexed for 1 min to remove the attached cells. One mL of sample was taken from each solution, serially diluted in buffered peptone water (BPW), and plated on Aerobic Plate Count PetrifilmTM in duplicate. Petrifilm was used in this study because the *Salmonella* strain was a pure culture (i.e., no other aerobic microorganisms were present), and the LWE did not show the presence of any aerobic bacteria. All Petrifilms were incubated at 37 °C for 24–48 h and counted manually.

### 2.5. Rapid Detection of Salmonella During Early Attachment

Two procedures were performed to identify *Salmonella* attached to coupons after 1 h of incubation, and both consisted of positive controls (i.e., LWE inoculated with *Salmonella*) and negative controls (i.e., LWE without inoculation). For the first procedure, the coupons were left in the culture plate, the LWE was aspirated, sterile saline solution (0.85% *w*/*v*) was used to rinse and remove any remaining LWE, and the surface of the coupon was swabbed. The swab was placed in 1 mL of phosphate-buffered saline (PBS) and swirled around the tube vigorously to release any bacteria. The tubes of PBS were centrifuged for 1 min at 13,000× *g*. The supernatant was removed, and DNA was extracted from the pellet using a Qiagen DNeasy PowerFood Microbial Kit (Qiagen, Germantown, MD, USA) according to the manufacturer’s instructions. DNA concentration and quality measurements were taken with a Denovix DS-11 FX+ spectrophotometer (DeNovix Inc., Wilmington, DE, USA).

For the second procedure, coupons were removed from the culture plates with sterile tweezers, transferred to 10 mL of sterile saline solution (0.85% *w*/*v*) for 1 min, and then placed in 5 mL of fresh saline solution containing glass beads. Samples were vortexed for 1 min, after which the beads were removed with sterile tweezers. The remaining liquid was centrifuged for 1 min at 13,000× *g*, the supernatant was removed, and DNA was extracted from the pellet with a Qiagen DNeasy PowerFood Microbial Kit. DNA concentration and quality were measured with a Denovix DS-11 FX+ spectrophotometer.

### 2.6. DNA Sequencing and Bioinformatics

Sequencing libraries were prepared using the Rapid Barcoding Kit (SQK-RBK114, Oxford Nanopore Technologies [ONT], Oxford, UK), according to the manufacturer’s instructions, which added sequencing adapters and unique barcodes to each sample. A total of 200 ng of DNA from each sample was used as input. Sequencing was performed on a GridION device (ONT) with R10.4.1 flow cells (ONT). Prior to sequencing, the flow cell was checked to ensure enough pores were available. Sequencing parameters in the MinKNOW software (version 24.02.6, ONT) were kept at default except for the following: 24 h run length, fast basecalling, a minimum quality score of 8, and a minimum read length of 1 kb.

The FastQ sequencing files were uploaded to the Galaxy platform for analysis [29]. Quality control was performed with Porechop (version 0.2.4) and Fastp (version 0.23.4) using a Phred quality cutoff of 8 and minimum length of 1 kb. Nanoplot (version 1.42.0) was used to assess the data after quality control. Reads that had passed quality control were analyzed with Kraken2 (version 2.1.1) with confidence at 0.5, *Salmonella* In Silico Typing Resource (sistr_cmd) (version 1.1.1), RefSeq Masher Contains (version 0.1.2) with a mash distance minimum identity to report set at 0.9, and SeqSero2 (version 1.2.1) with nanopore reads as input to identify any sequences from *Salmonella*. Kraken2 compares substrings of k length (k-mers) to determine the lowest common ancestor [30]. SISTR predicts the *Salmonella* serovar using genoserotyping and multilocus sequencing typing (MLST) [31]. SeqSero2 also predicts *Salmonella* serovar; however, it uses k-mers, not MLST [32]. Masher functions by clustering genomes and then mapping sequences to the appropriate reference to identify the taxon [33]. The FastQ files were also uploaded to the ONT platform EPI2ME and analyzed with What’s In My Pot (WIMP, version 2023.06.13-1865548) with the minimum length filter set at 1 kb.

### 2.7. Electron Microscopy

Coupons without any sample were observed under a Scanning Electron Microscope (SEM) to compare the attachment surface between the three materials. For these observations, no sample preparation was required. Coupons were mounted on stubes and gold-coated (EMS 150R, EM Sciences, Hatfield, PA, USA). A FEI Quanta 200 F Scanning Electron Microscope (Hillsboro, OR, USA) was used to see the samples in high vacuum mode (accelerating voltage, 10 kV). Additional coupons at incubation times 0 and 5 h were prepared using the same conditions as described in Section 2.3 for SEM observation. Excess LWE was aspirated from each coupon, rinsed with 2 mL of sterile saline solution (0.85% *w*/*v*), and replaced with 2 mL of 2% glutaraldehyde overnight. Sample preparation for cell attachment and biofilm formation on coupons for SEM has been previously described by Bermudez-Aguirre et al. (2025) [28].

### 2.8. Statistical Analysis

All experiments were conducted at least in duplicate on different days with a new and fresh inoculum. Basic statistical analyses (i.e., average and standard deviation) were conducted using Microsoft Excel (Version 2008, Seattle, WA, USA). A one-way analysis of variance was conducted using Excel to determine any significant difference (α = 0.05) between the food-contact surfaces studied in the present work.

## 3. Results

### 3.1. Culturing

Attachment of *Salmonella* suspended in LWE to coupons was quantified using conventional plating techniques. There was no growth of *Salmonella* at the 0, 1, or 2 h time points on any of the coupons. After 3 h of incubation, cell attachment was detected at less than 1 log CFU/cm^2^ of *Salmonella*, as shown in Figure 1. The material with the most *Salmonella* attached was silicone, followed by stainless steel. Nylon had the fewest *Salmonella* attached, with approximately 0.5 log CFU/cm^2^. After 4 h of incubation, attachment was very similar between the three materials, with growth of about 1.5 log CFU/cm^2^ of *Salmonella*. At the last time point (5 h), all tested materials had more than 2 log CFU/cm^2^ of attached *Salmonella,* with the highest level of attachment on stainless steel. However, there were no significant differences (*p* > 0.05) between the amount of *Salmonella* attached to stainless steel, nylon, or silicone at any of the time points. There was no growth in the negative controls at any time point.

### 3.2. Long-Read Sequencing

The inability to detect *Salmonella* on coupons prior to 3 h of incubation with culturing techniques prompted the use of sequencing to determine if it could detect *Salmonella* at the 1 h time point. Table 1 and Table 2 provide the metrics of the sequencing runs. Several bioinformatic tools were used to identify any *Salmonella* DNA in the samples. SeqSero2, SISTR, and RefSeq Masher Contains did not identify DNA from *Salmonella* in any of the samples. However, Kraken2 and WIMP detected *Salmonella* sequences on all three surface materials—stainless steel, silicone, and nylon—when they were swabbed in both experiments (Table 1 and Table 2). When bead processing was used instead of swabbing, only the nylon coupons were positive for *Salmonella* in the first experiment (Table 1). No *Salmonella* was detected in any of the negative controls (Table 1 and Table 2).

### 3.3. Scanning Electron Microscopy

The SEM images of inoculated and uninoculated coupons are shown in Figure 2. The uninoculated stainless steel (Figure 2a), silicone (Figure 2b), and nylon (Figure 2c) contact surfaces were free of any biological material. After inoculation of the coupons with *Salmonella* in LWE, SEM images were obtained at the 0 h time point, and a protein matrix was visible on all coupon materials, but no *Salmonella* cells were observed (Figure 2d–f). At the 5 h time point of incubation, microbial cells were difficult to identify within the complex topography of the incipient formation of a biofilm (Figure 2g–i).

## 4. Discussion

Conventional plating methods are typically used to identify and quantify microorganisms in different food products and on food-contact surfaces. In this study, plating was able to detect *Salmonella* attachment to all three coupon materials—stainless steel, silicone, and nylon—starting at the 3 h time point, and it increased correspondingly with time. However, no *Salmonella* could be cultured from the coupon surfaces at the 1 and 2 h time points. Therefore, samples from the 1 h time point were tested with long-read sequencing to determine if it could be a more sensitive detection method. *Salmonella* DNA was detected on all three surfaces, suggesting that long-read sequencing could be useful for early detection of *Salmonella* attachment to surfaces. In addition to its sensitivity, long-read sequencing has other advantages when used for testing. Multiple samples (up to 96) can be barcoded differently and run on the same sequencing flow cell, saving labor and materials in comparison with culturing. Results can also be obtained in about 24 h, similar to plating. Additionally, sequencing can provide information on serotype, antibiotic resistance genes, and virulence genes. It is also non-specific and can detect any microorganisms present, while culture-based methods require multiple selective medias for different bacteria. A disadvantage of sequencing is that it is not quantitative. Nonetheless, if only detection is required, this would not be a hindrance. Long-read sequencing has the potential to reduce the time and expense needed to test food processing equipment for contamination.

Two methods to remove *Salmonella* from coupon surfaces were tested. Bead processing was selected because it is the method typically used in our laboratory for enumerating *Salmonella* in biofilms with culturing [28]. However, it was hypothesized that very small numbers of *Salmonella* would be attached to the coupon at the 1 h time point. Therefore, swabbing was also tested because the contents of the swab could be released into a smaller volume of liquid for subsequent concentration. *Salmonella* was detected on all coupon materials with sequencing when the surface was swabbed, and DNA was extracted from the collected material, but only nylon coupons tested positive for *Salmonella* when bead processing was used. DNA concentrations extracted from all samples were generally low below 10 ng µL^−1^. However, the average concentration obtained from bead-processed samples was lower (1.948 ng µL^−1)^ than swabbed samples (4.762 ng µL^−1)^. The lower DNA concentration obtained from the bead-processed samples likely led to a lack of detection in the stainless steel and silicone coupons. Previous research with *Escherichia coli* O157:H7 demonstrated that sequences from *E. coli* could be detected in DNA concentrations as low as 0.39 ng µL^−1;^ however, at least 12.5 ng µL^−1^ of DNA had to be sequenced to identify the genes required to serotype the *E. coli* as O157:H7 [3]. To classify reads as belonging to a particular species, both Kraken2 [34] and WIMP [35] identify exact matches of k-mers in the sequencing reads to k-mers in a database of existing genomes. Less sequencing data as a result of low DNA concentrations would reduce the chance of finding the sequences needed to identify bacteria.

Sequences from *Salmonella* were also only identified in the coupon DNA extractions using the bioinformatic tools Kraken2 and WIMP but not with SeqSero2, SISTR, or RefSeq Masher Contains. This may be due to differences in how the programs analyze the data. As stated in the previous paragraph, Kraken2 [34] and WIMP [35] compare k-mers in the sequencing reads to k-mers in genomic databases to determine what species a particular read originates from. However, SISTR uses genoserotyping and MLST [31], Masher maps sequences to references [33], and SeqSero2 identifies genetic determinants [32]. The number of reads generated was low (Table 1 and Table 2) due to the low concentration of DNA extracted from the samples. The k-mer-based tools were able to identify *Salmonella* in the available sequences, but there may not have been enough data for the tools that use other approaches to classify the reads.

Biofilm formation involves multiple steps and is considered a cyclic process. During the first step, known as reversible attachment, the planktonic cells are initially associated with the surface in a transient way. After a few hours, the cells move to the second stage, irreversible attachment. During the third stage, called maturation I, the cells start producing EPS and form cell clusters on the surface. Maturation II is the fourth stage when microcolonies form, and finally, the last stage is known as dispersion [36,37]. In the present research, culturing results for the last time point, after 5 h of incubation, indicated there was over 2 log CFU/cm^2^ of *Salmonella* attached to the coupons. These attached cells do not represent a mature biofilm, though they are the foundation of a biofilm that can mature with longer incubation time. The same *Salmonella* strain used for the present research was studied for biofilm formation after 24 and 48 h in a previous study, and the results showed a strong biofilm was present after 24 h, with more cells attached to nylon than silicone or stainless steel [28]. The biofilm architecture consists of extracellular materials that are difficult for sanitizers to penetrate to kill the bacteria within [38]. The ability of *Salmonella* to form biofilms on various surfaces in egg processing facilities needs to be considered when selecting sanitization interventions to ensure effective removal. Currently, there is extensive research being conducted to find alternatives for chemical sanitizers such as chlorine, quaternary ammonium compounds, and hydrogen peroxides. These chemicals can generate toxic by-products and contribute to the development of antibiotic resistance of bacteria [39]. Natural sanitizers obtained from vegetable sources have been successfully used as sanitizers on food-contact surfaces against *Salmonella* [39,40]. However, further research is required to validate these results and ensure the sanitization of the food- contact surfaces. Several references have shown that the longer the biofilm forms, the stronger the intervention needed for removal [5,6,8,18]. The ability to detect biofilm formation earlier with long-read sequencing would allow remediation to occur sooner and prevent the formation of a robust biofilm.

In this study, there were no significant differences in adherence to the three contact surfaces: stainless steel, silicone, and nylon. The highest numbers of *Salmonella* were attached to silicone at the 3 h time point (Figure 1), which is the first time point when *Salmonella* could be cultured. However, at the 5 h time point, more *Salmonella* were adhered to stainless steel than silicone or nylon. Other studies have also noted variable attachment by *Salmonella* to different materials [14,41,42]. It has been suggested that the characteristics of a material, such as roughness, hydrophobicity, contact angle, and streaming potential, affect bacterial attachment [43]. One study showed that *S.* Typhimurium ATCC 14028 attached more strongly to stainless steel than acrylic, rubber, and polyurethane [44]. The authors suggested that *Salmonella* formed stronger biofilms on stainless steel because it is a less hydrophobic material than plastic. Another study found that three *Salmonella* strains (*S. enteritidis*, *S. agona*, and *S. newport*) adhered more strongly to nylon than wood or high-density polyethylene [14]. Yet another experiment demonstrated that *S. enteritidis* was more challenging to remove from silicone rubber than stainless steel using common sanitizers such as chlorine-based compounds due to the strong biofilm formed by the cells [45]. In addition to differential attachment to materials, the *Salmonella* serovar also plays an important role in adherence strength. Oliveira et al. (2007) evaluated the elemental composition and cell surface physicochemical properties of four strains of *S.* Enteritidis when attaching to stainless steel, but these properties did not influence the strength of attachment [41]. The authors suggested that other serovar-specific characteristics may be playing a role. The amount of time *Salmonella* has to attach [14] and the environmental temperature [42] also influence attachment strength to materials. The results of this study and previous work by other researchers reinforce the need to detect *Salmonella* attachment to any surface as early as possible to perform remediation.

The images obtained through electron microscopy showed differences in the surfaces of the three materials. Stainless steel 304 L is used widely in the food industry and considered “food grade” because of its resistance to corrosion and hygienic properties [46]. However, some of the properties of stainless steel such as the grade of steel, type of finish, the surface roughness, and age, can impact its performance and cell adherence [47]. Figure 2a shows a stainless steel surface with irregular lines that can easily harbor the growth of microorganisms and promote the formation of biofilms. It has been reported that the presence of cracks and deformations on the material surface increases the adhesion of cells and makes removal more difficult; therefore, the roughness of the material should be considered for food-contact materials [43]. The SEM images of cell attachment in the first hours of incubation (Figure 2d–i) present a complex structure of fat and protein characteristic of liquid whole egg. This rich and nutritive medium promotes the growth of microorganisms. *Salmonella* cells are difficult to identify at the 0 h time point because the number adhered is very low. At the 5 h time point, *Salmonella* cannot be seen due to the complex matrix present. According to the plate count, at 5 h, stainless steel had the highest cell attachment, with about 3 log CFU/cm^2^. These images show the incipient formation of a strong protein biofilm that will mature with the multiplication of *Salmonella* cells and the production of EPS. An effective intervention should be applied to these food-contact surfaces to remove the organic matter (i.e., liquid egg) and inactivate the *Salmonella*. Some common sanitizers, like chlorine solutions, lose effectiveness when organic matter is present [48,49]. Therefore, it is critical to detect pathogens on food-contact surfaces early and to find effective interventions for young biofilm removal.

## 5. Conclusions

Current food safety challenges in the food industry are not only related to the control of pathogens in food and processing areas but also to the rapid detection of pathogens. *Salmonella* present in liquid egg can easily attach to stainless steel, silicone, and nylon after 1 h of contact. However, conventional detection methods like plating are not only lengthy but also limited in their ability to detect cells in the early stages of contamination. Sequence methods offer a novel and fast option to quantify bacterial DNA in liquid egg. This innovative detection method was able to identify the presence of *Salmonella* in the early stages of cell attachment. Additional optimization of sample collection and DNA extraction methods will improve *Salmonella* read detection with bioinformatic tools. Further experimentation simulating real-case scenarios with background flora should be investigated and validated before being transferred to the food industry.

## Figures and Tables

**Figure 1 microorganisms-13-00548-f001:**
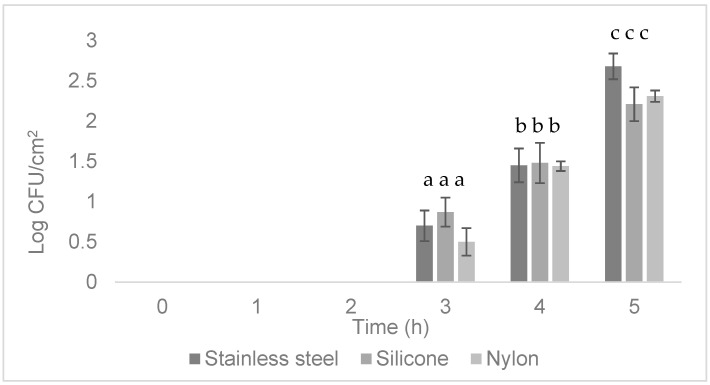
Attachment of *Salmonella* Typhimurium 53,647 to three different surfaces (stainless steel, silicone, and nylon) during the first five hours of incubation (37 °C, static conditions) using conventional microbiological methods. a, b, c: The same letters at a time point indicate no significant difference in attachment between materials (*p* > 0.05).

**Figure 2 microorganisms-13-00548-f002:**
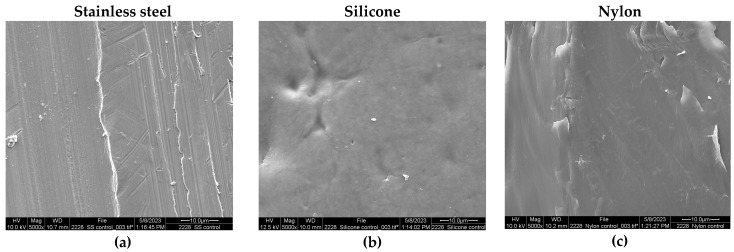
Scanning electron microscopy images of the clean food-contact surfaces materials: (**a**) stainless steel, (**b**) silicone, (**c**) nylon. Liquid whole egg inoculated with *Salmonella* at time 0 h: (**d**) stainless steel, (**e**) silicone, (**f**) nylon. Cell attachment samples at time 5 h: (**g**) stainless steel, (**h**) silicone, (**i**) nylon. All images are at 5000× magnification.

**Table 1 microorganisms-13-00548-t001:** Sequencing metrics from the first experiment.

Sample Description	Mean Length (b)	Length Standard Deviation (b)	Estimated Bases (Mb)	Reads Generated	Mean PHRED Score	N50 (b)	WIMP	Kraken2
NC Stainless Steel	2562.8	1558.8	52.8420	20,619	10.8	2971.0		
NC Silicone	3037.8	2069.5	91.7236	30,194	10.8	3748.0		
NC Nylon	3097.5	2051.6	76.0129	24,540	10.8	3812.0		
Stainless Steel	2752.6	1817.3	33.9995	12,352	10.8	3296.0	+	+
Silicone	2351.2	1418.6	55.3644	23,547	10.8	2682.0	+	+
Nylon	3011.0	1942.1	68.1127	22,621	10.8	3667.0	+	+
NC Wash Stainless Steel	2658.3	1440.8	0.0266	10	10.2	3535.0		
NC Wash Silicone	2293.3	1806.2	0.0871	38	11.3	2821.0		
NC Wash Nylon	2392.0	1182.9	0.0574	24	10.5	2759.0		
Wash Stainless Steel	1786.1	663.8	0.0125	7	11.4	2087.0		
Wash Silicone	2085.8	1010.4	0.1043	50	10.5	2314.0		
Wash Nylon	2486.3	2176.7	0.0696	28	10.9	2894.0	+	

NC = Negative control; Wash = Samples processed with bead beating.

**Table 2 microorganisms-13-00548-t002:** Sequencing metrics from the second experiment.

Sample Description	Mean Length (b)	Length Standard Deviation (b)	Estimated Bases (Mb)	Reads Generated	Mean PHRED Score	N50 (b)	WIMP	Kraken2
NC Stainless Steel	3158.0	1092.0	0.0063	2	11.5	4250.0		
NC Silicone	1335.0	358.5	0.0093	7	10.5	1279.0		
NC Nylon	1210.0	0.0	0.0012	1	10.0	1210.0		
Stainless Steel	1247.0	66.3	0.0037	3	9.9	1283.0	+	+
Silicone	3659.3	32.6	0.0110	3	9.6	3642.0	+	+
Nylon	17,083.9	55,646.2	0.5638	33	9.1	315,569.0	+	+
NC Wash Stainless Steel	94,082.5	92,959.5	0.1882	2	8.0	187,042.0		
NC Wash Silicone				0				
NC Wash Nylon	425,978.0	0.0	0.4260	1	15.0	425,978.0		
Wash Stainless Steel	1192.0	0.0	0.0012	1	9.0	1192.0		
Wash Silicone	8533.0	0.0	0.0085	1	8.0	8533.0		
Wash Nylon	1720.5	188.5	0.0034	2	11.0	1909.0		

NC = Negative control; Wash = Samples processed with bead beating.

## Data Availability

The data presented in this study are openly available in USDA at https://doi.org/10.15482/USDA.ADC/26866363 [50].

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
