# Peer review of "Rapid Detection of Salmonella Typhimurium During Cell Attachment on Three Food-Contact Surfaces Using Long-Read Sequencing"

_microorganisms, 2025, doi:10.3390/microorganisms13030548_

Round 1

Reviewer 1 Report

Comments and Suggestions for Authors

The objectives of this study were very clearly defined. The manuscript was scientifically written, with a logical design. Methods were well-described for repeatability. Also, results were discussed reasonably with other reports. The authors used liquid whole eggs for cell attachment to prepare the surface samples, which was very important to mimic the actual practice. More importantly, the finding was very useful for the food industry. However, I had a few questions to include in this or further study.

1.      Why didn’t you inoculate background microorganisms to verify the detection methods?

2.      Why did you choose glass beading to recover Salmonella from surfaces? Swabbing is more practical if you have a large area.

Line 44, “have” should be “has.”

Line 46, this study was to detect Salmonella on food-contact surfaces, but biofilm formation on eggshells was an irrelevant example to this study. Please change to another example with a reference.

Line 48-51, these two sentences didn’t flow well. I suggested combining two sentences: "The materials of food-contact surfaces, including stainless steel, glass, silicone, polyurethane, Teflon and wood, influences…”

Line 171-174, could you please include more details about Kraken2, Sistr, RefSeq Masher Contains, seqsero2? Some readers are not experts in using these systems and might need to know the function of those.

Line 293, was this comparison based on Figure 1? Have you used the long-read sequencing method to detect 2-h samples, as the conventional method still could not detect Salmonella?

Line 310-313, this was in conflict with your previous results, as you didn’t observe significant differences in adherence. If any previous study supported this statement, please add the reference here.

Author Response

Reviewer #1

The objectives of this study were very clearly defined. The manuscript was scientifically written, with a logical design. Methods were well-described for repeatability. Also, results were discussed reasonably with other reports. The authors used liquid whole eggs for cell attachment to prepare the surface samples, which was very important to mimic the actual practice. More importantly, the finding was very useful for the food industry. However, I had a few questions to include in this or further study.

We are very thankful for your comments and suggestions to improve our manuscript. Please find below the answers to some questions and the corrections in the manuscript.

Comment 1. Why didn’t you inoculate background microorganisms to verify the detection methods?

Response 1. Thank you for your comment. This manuscript represents the first approach with a fast detection method in egg biofilms. Since Salmonella is the main pathogen of concern in the egg industry, we decided to address the pathogen itself in the biofilms with this novel detection method. As our project is progressing, we are currently testing the presence of Pseudomonas as background flora in the biofilm formation, alone and in combination with Salmonella. Once we find the best biofilm formation conditions for these microorganisms, we will evaluate the use of novel detection techniques for both bacteria.

Comment 2. Why did you choose glass beading to recover Salmonella from surfaces? Swabbing is more practical if you have a large area.

Response 2. Thank you for your comment. We used both for recovery of Salmonella. For biofilm quantification we used glass beading to remove the biofilm from the surfaces, and for the fast detection method we swabbed the surface. The use of glass beading has been a standard practice in our lab during the study of biofilms with accurate results.  

Comment 3. Line 44, “have” should be “has.”

Response 3. Thank you for your comment. The correction has been made.

Comment 4. Line 46, this study was to detect Salmonella on food-contact surfaces, but biofilm formation on eggshells was an irrelevant example to this study. Please change to another example with a reference.

Response 4. Thank you for your comment. Certainly, biofilms on eggshells are not the main topic here. We changed the sentence, because our intention is to mention the problem of cross-contamination of eggshells when the egg enters in contact with contaminated surfaces that have biofilms.

Comment 5. Line 48-51, these two sentences didn’t flow well. I suggested combining two sentences: "The materials of food-contact surfaces, including stainless steel, glass, silicone, polyurethane, Teflon and wood, influences…”

Response 5. Thank you for your comment. We have changed the sentences in the manuscript for easier reading.

Comment 6. Line 171-174, could you please include more details about Kraken2, Sistr, RefSeq Masher Contains, seqsero2? Some readers are not experts in using these systems and might need to know the function of those.

Response 6. Thank you for the suggestion. Additional details were provided about each of the bioinformatic programs.

Comment 7. Line 293, was this comparison based on Figure 1? Have you used the long-read sequencing method to detect 2-h samples, as the conventional method still could not detect Salmonella?

Response 7. Thank you for your comment. Yes, this comparison was based on Figure 1. We did not use sequencing on 2h samples since we were able to detect the Salmonella on the coupons in the earlier 1h time point.

Comment 8. Line 310-313, this was in conflict with your previous results, as you didn’t observe significant differences in adherence. If any previous study supported this statement, please add the reference here.

Response 8. Thank you for your comment. We added a couple of references to support this statement.

Reviewer 2 Report

Comments and Suggestions for Authors

Manuscript 3466945

Journal Microorganisms

Title Rapid detection of Salmonella Typhimurium during cell attachment on three food-contact surfaces using long-read sequencing

The manuscript entitled “Rapid detection of Salmonella Typhimurium during cell attachment on three food-contact surfaces using long-read sequencing” describes the detection of Salmonella spp. on three food-contact surfaces by plate counting or long-read whole genome sequencing. Two protocols for the detachment of cells were compared.The topic is interesting but several parts need revision. Please follow the comments in the file.

Author Response

The manuscript entitled “Rapid detection of Salmonella Typhimurium during cell attachment on three food’s contact surfaces using long-read sequencing” describes the detection of Salmonella spp. on three food’s contact surfaces by plate counting or long-read whole genome sequencing. Two protocols for the detachment of cells were compared. The topic is interesting but several parts need revision.  

We are very thankful for your comments and suggestions to improve our manuscript. Please find below the answers to some questions and the corrections in the manuscript.

Comment 1. L12-26 Revise the abstract including quantitative data

Response 1. Thank you for your comment. We added quantitative data to the abstract.

Comment 2. L37-40 Please include the survival rate on different food-processing surfaces, and the biofilm-forming ability on different food-contact surface. Expand this part

Response 2. Thank you for your comment. We added more information about this and expanded the paragraphs.

Comment 3. L47-49 Please be more specific. How do these parameters affect adhesion and biofilm formation?

Response 3. Thanks for your comment. A new paragraph has been added to the text regarding this information.

 Comment 4. L52-53 Revise this part. Please include also the removal of planktonic cells and the use of essential oils. The papers doi.org/10.1016/j.ifset.2018.02.013 and doi.org/10.3390/antibiotics13040371 are suggested for your analysis and discussion.

Response 4. Thank you for your comment. We added these references to our manuscript and briefly discussed the use of these natural sanitizers for food contact surfaces.

Comment 5. L58-65 Compare these methods by using the time to results, sensitivity, selectivity and real application to detect planktonic cells and biofilms of Salmonella. Revise this part

Response 5. Thank you for the suggestion. The requested information was added to the introduction.

 Comment 6. L66-72 Are there applications related to foodborne pathogens detection on food contact surfaces? Please add specific examples

Response 6. Thank you for the question.  No, long-read sequencing has not been used for direct detection of foodborne pathogens on food contact surfaces. A sentence was added to this part of the introduction to clarify that point.

Comment 7. L114 How the inoculum cell density was selected?

Response 7. Thank you for your question. The inoculum cell density was selected based on previous experiments (it has been added to the text). The aim of this research was to simulate real-case-scenario conditions to use these results in the food industry after optimization.

Comment 8. L124-125 Why sequencing was done only at 1 h? Please explain

Response 8. Thank you for your question. Sequencing was done at 1 h because it really represents the early stage of cell attachment. The aim of this new detection method is to detect the cells at early stages of attachment to the surface. If cells are not detected earlier, then the biofilm gets thicker and cells are attached stronger, representing a problem not only about cross-contamination with foods, but also in the removal of the biofilm and sanitization of the food contact surface.

Comment 9. L127-136 Why the authors did not evaluate the Salmonella in the remaining LWE? It is important to evaluate the attachment of cells to food contact surfaces

Response 9. Thank you for your question. In this specific case, we focused on the cell attachment in the different surfaces rather than the remaining cells in the LWE.

Comment 10. L195-196 Please add the software and the post-hoc test applied to differentiate mean values

Response 10. Thank you for your comment. The information has been added to the text.

Comment 11. L153-154 Why two washing steps were considered?

Response 11. Thank you for your question. We tested the bead beating wash because that is the typical preparation method that has been used for biofilm samples prior to culturing. We tested the swabbing method because that is a common method for collecting samples for sequencing. We thought it might be easier to concentrate the bacteria in a smaller volume of liquid if they were collected on a swab.

Comment 12. L160-166 Add more details on the sequencing (e.g., preparation of the library, sequencing conditions and so on). Revise this part

Response 12. Thank you for the suggestion. Additional details on library preparation and sequencing were added to the methods.

Comment 13. L199-210 and Figure 1 Please report the data in log cfu cm-2 . Revise the figure and the text

Response 13. Thank you for your comment. The figure and text have been updated.

Comment 14. L220-224 Why these discrepancies? Please add a possible explanation

Response 14. Thank you for your question. Possible explanations for the discrepancies were added to the discussion.

Comment 15. L224-225 Why? Please add a possible explanation

Response 15. Thank you for your question. A possible explanation was proposed in the discussion.

Comment 16. L240-243 Why images were not acquired from 0 to 5 h (e.g., 2 h)?

Response 16. Thank you for your question. We have done several experiments related to eggs and biofilms using electron microscopy at the different stages. Based on our experience, we wanted to show the images with scientific value. The images at shorter processing times (<5 h) do not show clearly the cells because of low counts and the big interference due to the egg matrix. So, we decided to include in the manuscript the images that shows a difference between the initial and final stages of the cell attachment in the different materials.

Comment 17. L269-271 Is the optimization of the sample preparation necessary? Please discuss this aspect

Response 17. Thank you for your question. A discussion of the need to optimize sample preparation was added.

Comment 18. L279-280 Discuss the result related to the attachment on different food-contact surfaces

Response 18. Thank you for your comment. We discussed in detail this section.

Comment 19. L300-307 Add more references related to the discussion of these results

Response 19. Thank you for your comment. We added more references and discussed in detail this section.

Comment 20. L328-329 Add a conclusion section

Response 20. Thank you for the suggestion. A conclusion has been added to the manuscript.

Round 2

Reviewer 2 Report

Comments and Suggestions for Authors

Authors addressed reviewer's comments. Minor comments are reported below:

1 Add in the conclusion section the need to optimize DNA extraction to find the reads of Salmonella with different bioinformatic tools

2 Revise the English throughout the manuscript. Some sentences are difficult to read and understand

Comments on the Quality of English Language

English should be revised

Author Response

Thank you for taking the time to review the manuscript again and provide additional comments. Our responses are below.

Comment 1. Add in the conclusion section the need to optimize DNA extraction to find the reads of Salmonella with different bioinformatic tools

Response 1. The need to optimize DNA extractions to improve bioinformatic detection was added to the conclusion.

Comment 2. Revise the English throughout the manuscript. Some sentences are difficult to read and understand

Response 2. The entire manuscript was reviewed for grammar and clarity and corrections made.